# Understanding the impact of urban exposure on obesity among middle and old-age migrants in India

**Bittu Mandal** *, **Kalandi Charan Pradhan**

School of Humanities and Social Sciences, Indian Institute of Technology Indore, Simrol, Indore, India

* mbittu545@gmail.com

## Abstract

Rural-to-urban migration is associated with elevated obesity in Western settings. However, whether migration to urban areas ages has any impact on obesity in India is inconclusive and scarce. We, therefore, assessed the impact of migration on obesity among rural-to-urban migrants and compared it with their rural counterparts. This study utilized the first wave of Longitudinal Ageing Study in India. BMI (BMI ≥ 25 kg/m$^2$) and waist circumference (>102 cm and >88 cm for men and women, respectively) were employed to assess overall obesity and abdominal obesity. To fulfil the study objective, this study employed logistic and quantile regression techniques. The study found that individuals migrating from rural to urban areas are significantly more likely to develop obesity than rural stayers. Moreover, within the group of rural-urban migrants, prolonged urban residence was a strong and cumulative predictor for obesity. The risk of obesity was 1.91 times higher (those who lived 5 or fewer years in urban areas), 2.05 times higher (for 6–10 years), and 2.40 times higher (for more than 10 years) compared to their rural counterparts. This study identified migration and prolonged urban exposure as crucial risk factors for the development of obesity among middle-aged and older adults in India.

## Introduction

The prevalence of obesity in adults has been continuously rising since the 1970s [1]. In 2016, the global prevalence of adult obesity reached 13.1%, marking an increase from 8.7% in 2000. The most significant rises were observed in Southeast Asia, where the prevalence surged from 1.9% to 4.7%, indicating an almost 150% increase [2]. Obesity is increasingly acknowledged as a disease in itself because of the elevated risk of morbidity and mortality [3]. Additionally, obesity is also closely linked to a number of other metabolic and other diseases, such as diabetes mellitus [4], hypertension [5], musculoskeletal conditions [6], mental illnesses [7], multiple cancers [8] and overall poor quality of life [9]. Not only does obesity contribute to health issues,

**Data availability statement:** The data are available in the public domain and freely accessible upon registrtion from the Gateway to Global Aging Data - www.g2aging.org Replicable data is uploaded in figshare. https://doi.org/10.6084/m9.figshare.27266337.v1.

**Funding:** The author(s) received no specific funding for this work.

**Competing interests:** The authors have declared that no competing interests exist.

but it also imposes a considerable health burden, leading to significant healthcare costs [10].

Obesity is influenced by both genetic [11] and environmental factors [12]. Apart from the genetic susceptibility to central obesity among South Asian descendants [11], numerous external factors influencing obesity have changed in recent years. Rapid changes in dietary habits, for example, can lead to an unhealthy nutritional transition characterized by the increase in fast food and processed foods with high energy, sweetened beverages, and salt intake [13,14]. Sedentary lifestyles include less physical activity, less cycling, and less walking for commuting [15]. The urban built environment, with its few open green spaces and reliance on passive transportation, also impacts obesity [16]. All these are primarily attributed to changes in lifestyle associated with urbanization, along with the migration from rural to urban areas and the impacts of globalization [4,17]. Between 1991 and 2011, the urban population in India expanded significantly, rising from 11% to 34%, encompassing 380 million people. Roughly 20% of this urban population growth is linked to individuals relocating from rural to urban regions [18]. Rural-to-urban migration has received considerable study focus as a potential factor in a variety of health problems, including both communicable and non-communicable diseases [19,20]. The risk of developing diseases like hypertension [5,21], diabetes [22], different cardiovascular diseases [23], cognitive impairment [24], and overall poor health [25] is significantly higher migrant populations. Many established research findings indicate that individuals migrating from rural to urban areas in low- and middle-income countries (LMICs) face an increased risk of obesity [4,26–28]. Additionally, prior studies have explored the connection between obesity and migration, particularly in relation to the duration of residence in urban areas [29,30]. Some empirical investigations have also delved into the impact of age at migration on the obesity risk within migrant populations [30,31]. Najera and colleagues illustrated that migrants from the high socio-economic strata were more prone to obesity [32]. Another study conducted by Murphy et al. (2017), showed a complex connection between neighbourhood deprivation and obesity risk among migrants, underscoring the significance of mediators. Furthermore, Kinra et al. (2020) identified an independent negative association between physical activity and obesity among Indian migrants. Elevated body weight in migrants was linked to negative life events and stress [28]. Recent research (Johnson et al., 2022) reported that a shorter duration of sleep is associated with a higher prevalence of obesity [33]. Another study from France found that migrants facing functional limitations were at a higher risk of developing obesity [34].

There is a limited body of research exploring the effects of rural-to-urban migration on obesity outcomes in India. In 2006−07, Ebrahim et al. (2010) conducted a migration survey in India, delving into key facets of rural-to-urban migration and its consequences. Nevertheless, this study had a limited sample size and focused on industrial workers lacked national inclusivity. Additionally, it did not focus on the middle-aged and elderly population. In the current scenario, in spite of the growing interest in this area, there have been few studies on the effect of rural-to-urban

migration and the length of residence on obesity, with a nationally representative sample of middle-aged and older adults. Most of the existing literature in this domain tends to concentrate on younger, working-age migrants or specifically on women. While these studies provide important insights, they leave a significant gap in understanding the obesity related health implications of migration among older adults. Middle-aged and older migrants represent a growing demographic in India's aging population, and they are particularly vulnerable to chronic conditions, including obesity, due to lifestyle changes post-migration, limited access to health services, and age-related metabolic shifts. Therefore, the primary objective of this study was to examine the hypothesis that rural-to-urban migration and length of residence in the urban areas is associated with an increased likelihood of obesity among Middle-aged and older rural-to-urban migrants. [19,20].

## Methods

### The data

The present study used data from the Longitudinal Aging Study in India (LASI) baseline wave (wave 1) conducted during 2017–2018 in India. The survey is a joint undertaking of the Harvard TH Chan School of Public Health, the International Institute for Population Sciences, and the University of Southern California. The nationally representative longitudinal survey collects vital information on the physical, social and cognitive wellbeing of India's older adults, which will be followed up for 25 years. The data of over 73000 individuals aged 45 and above, along with their spouses (irrespective of age), are collected across all states and union territories of India. The sample is based on a multistage stratified cluster sample design, including three and four distinct stages of rural and urban area selection, respectively. The survey provides scientific insights and facilitates a harmonized design, which helps compare with parallel international studies. Further, the details of sample design, survey instruments, fieldwork, data collection and processing, and response rates are publicly available in the LASI report and elsewhere [35–37]. The Indian Council of Medical Research provided the necessary guidance, guidelines, and ethical approval for conducting the survey. All methods were carried out in accordance with these relevant guidelines and regulations. Prior consent, both signed and oral, was obtained for the interviews and biomarker tests from the eligible respondents in accordance with the protection of human subjects. Participants of the study consist of 31595 middle aged and older adults aged 45 years and above in India. Detailed sample section procedure is reported in Figure S5 in S1 File.

### Variables

**Outcome variables.** Obesity-related measures were the outcome of interest for this study. Body mass index (BMI) is calculated by dividing an individual's weight (in kilograms) by the square of their height (in metres). A trained interviewer has collected the BMI information. Height was measured in centimetres using a stadiometer, and weight was measured in kilograms using a Seca 803 digital weighing scale. Based on the World Health Organisation (WHO) international cut-of, the BMIs of older Indian adults were classified as people with underweight (BMI ≤ 18.4), people with normal weight (BMI 18.5 to 24.9), people with overweight (BMI 25 to 29.9), and people with obesity (BMI ≥ 30) [38]. The respondents having a BMI of 30 and above were categorized as people with obesity and coded as "yes", otherwise "no" [9].

Waist and hip circumferences were measured in centimetres using a Gulick tape according to standard protocols. The critical threshold value for high-risk WC for men and women was ≥ 102 cm and ≥ 88 cm, respectively. Further, the high-risk waist-hip ratio was coded as no and yes [36,39].

Similarly, WHO categorizes waist-to-hip ratio (WHR) (waist circumference in cm/hip circumference in cm) into low and high-risk levels for men and women separately. The critical limit classification for the high-risk WHR for men was ≥ 0.90, and for women was ≥ 0.85 [36,39]. The respondents who had a high-risk WHR were coded as "yes", otherwise "no".

### Main explanatory variable

**Migration status.** Migration refers to a change of residence across defined geographic areas within a specified time interval, typically involving a move from one administrative boundary to another [40]. The Longitudinal Ageing Study in India also adopts the definition used by the Census of India, where individuals are classified as migrants if their place of enumeration during the survey differs from their last usual place of residence [41]. The concept of the *place of last residence* is crucial, as it captures the most recent relocation, including return migration [42]. Accordingly, this study defines a migrant as someone whose current residence differs from their last usual residence, indicating a permanent move across an administrative boundary. Several studies [43,44] have employed this method to determine migration status.

Finally, as our study is explicitly focused on the rural-to-urban migration, other migrants (rural-to-rural, urban-to-urban, and urban-to-rural) were excluded from the study [25]. To get the effect of urban exposure on migrants' health, rural-to-urban migrants were further classified into three categories based on the years of residence in urban areas- '0-5 years', '6-10 years', and 'more than 10 years' of migration [29,45,46].

### Covariates

The following variables were included in the analysis due to their association with outcome variables as previously reported in the literature like [4,47–51]: demographic, household and health behaviours, including age (45–59 and ≥ 60), sex of the respondents (male and female), educational attainment (no education, primary, secondary, and higher), employment status (currently working, never worked, and retired), wealth quantile (poorest, poorer, middle, richer and richest), caste (scheduled caste, scheduled tribe, other backward classes and others) religion (Hindu, Muslim and others), physical activity (active and inactive), alcohol consumption (never, infrequent, frequent, and heavy), tobacco consumption (non-consumer, and current smoker), difficulty in performing ADL (yes, no), mobility problem (yes, no), sleep problem (yes, no).

### Econometric analysis

Descriptive statistics, along with bivariate analysis, were utilized in the paper. The analysis was stratified by migration status. The prevalence of obesity was presented as proportions for each group stratified by individual, household, and behavioural factors. Further, a multivariable logistic regression model was performed separately in both sample groups to examine the adjusted effect of predictor variables on the likelihood probability of suffering from obesity.

$$Logit \ [P(Y=1)] = \beta_0 + \beta * X_n + \varepsilon_i \tag{1}$$

Where the parameter $\beta_0$ estimates the log odds of obesity for the reference group, while $\beta$ estimates the maximum likelihood, the differential log odds of obesity associated with a set of predictors $X_n$, as compared to the reference group and $\varepsilon_i$ represents the residual in the model.

In addition, quantile regression estimation was also employed to investigate the effect of migration status on the BMI distribution. Quantile regression is an extension of Ordinary Least Square (OLS) and accounts for the overall distribution of outcome measures based on a given percentile with a linear approach. To examine how the BMI is associated with the predictor, we estimate the following quantile regression.

$$Q_{yi|xi} \ (\tau|xi) = x_i^T \beta_\tau \tag{2}$$

Where $Q_{yi|xi} \ (\tau|xi)$ is the conditional $\tau^{th}$ quantile outcome $xi$, $\tau\epsilon(0,1)$ is the $\tau^{th}$ quantile of the BMI. For example, τ = [0.10, 0.20], and τ = 0.5 for median BMI regression. $xi = (xi_1, \ xi_2 \ldots xi_p)$, $T$ is the vector of covariates for each individual $i$, and $\beta_\tau = \beta_\tau 0, \ \beta_\tau 1 \ldots \beta_\tau$ T is the vector of regression coefficients at a known τ. There was no evidence of multicollinearity among the variables when the variance inflation factor was calculated using stata v.18 [52]. Additionally, individual weights were employed to make the results nationally representative.

# Results

The kernel density estimates of BMI score for rural non-migrant, urban non-migrant and rural-to-urban migrants (Fig 1). Urban non-migrants and rural-to-urban migrants had similar BMI profiles. BMI scores were skewed towards the right, indicating that these two groups had an elevated BMI compared to rural non-migrants.

Fig 2 provides the sex stratified weighted prevalence of obesity for the rural and urban non-migrants and rural-to-urban migrants based on their duration in the urban areas. Overall, rural non-migrants had the lowest prevalence of obesity (2.61%), whereas urban non-migrants had the highest prevalence of obesity (13.19%). An increasing trend of obesity

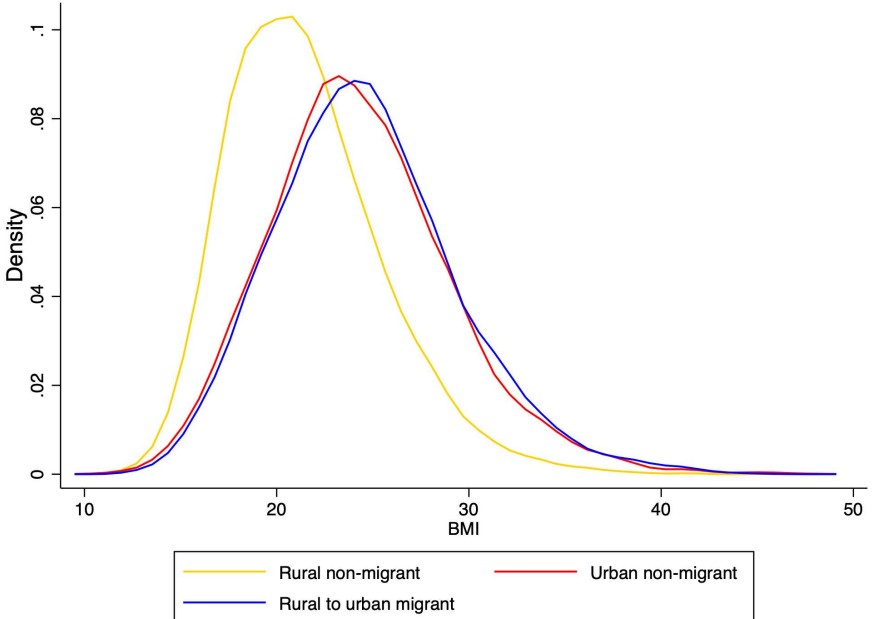

**Fig 1. Distribution of BMI score stratified by migration status among the middle aged and older adults (aged 45+) in India, LASI wave 1 (2017−18).**

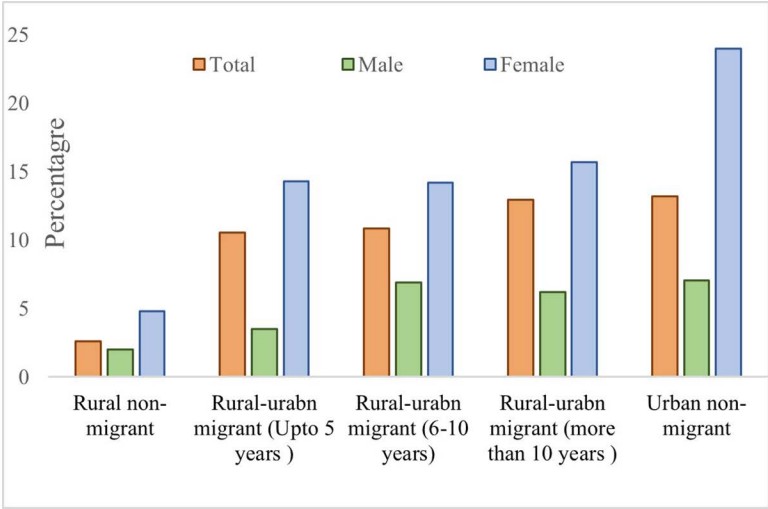

**Fig 2. Sex stratified weighted prevalence of obesity by residence status (non-migrants) and duration of residence in urban areas (among migrants) aged 45+ in India, LASI Wave 1 (2017−18).**

can be found in rural-to-urban migrants. As the duration of residence in the urban areas increases among the rural-to-urban migrants, the prevalence of obesity also steadily increases. Females demonstrated an elevated prevalence of obesity compared to males across the migrant and non-migrant groups.

Table 1 shows that among the respondents, 55.63% were rural non-migrants, 19.64% were urban non-migrants, and 24.73% were rural-to-urban migrants. 27.51% of rural non-migrants, 39.76% of the urban non-migrants and 64.56% of the rural-to-urban migrants were female. 49.46%, 25.65% and 35.23% of the rural non-migrant, urban non-migrant and rural-to-urban migrants had no formal education, respectively. About 20.83% of rural non-migrants, 24.83% of the urban non-migrants, and 21.09% of the rural-to-urban migrants belonged to the poorest economic status.

Table 2 shows the prevalence of obesity among the middle-aged and older population with respect to their migration status and background characteristics. Overall, 2.59% of the rural non-migrants, 13.19% of the urban non-migrants and 13.08% of rural-to-urban migrants were found to be with obesity. Among the rural-to-urban migrants, there was a higher prevalence of obesity among middle-aged individuals (15.70%) compared to older adults (9.48%). Rural non-migrants with a higher education (5.40%), urban non-migrants (19.36%), and rural-to-urban migrants (16.43%) with a secondary education reported higher obesity. Both in migrants and non-migrants, respondents who never worked and belonged to the highest economic strata reported a very high prevalence of obesity. Among the rural non-migrant, urban non-migrant, and rural-to-urban migrant respondents, those with a problem in performing ADL were 4.66%, 13.40%, and 13.91% with obesity, respectively. Additionally, those with mobility impairment were 3.11%, 16.81%, and 14.36% with obesity, respectively.

Table 3 shows the estimates of adjusted odds ratios for obesity and other socio-economic and demographic covariates from the multivariate logistic regression model. It shows the estimates of adjusted odds ratios for obesity separately for rural non-migrants, urban non-migrants, and rural-to-urban migrants. For the rural non-migrants, the likelihood of obesity was 2.86 times higher for females than males [OR: 2.86; CI: 2.07, 3.96]. In reference to the illiterate rural non-migrants, those who were highly educated, 3.45 times more likely to suffer from obesity [OR: 3.45; CI: 2.23, 5.36]. In comparison with the richest strata, those who were in the poorer [OR: 1.93; CI: 1.20, 3.09], middle [OR:3.15; CI: 1.97, 5.04] and richer [OR: 3.16; CI: 2.02, 4.91] economic strata, were more likely to have develop obesity. Respondents with mobility impairment [OR: 1.52; CI: 1.16, 1.99] and difficulty in ADL [OR: 1.57; CI: 1.01, 2.43] were more likely to have obesity than their counterparts without any such impairments.

Among the urban non-migrant respondents, females [OR: 2.08; CI: 1.50, 2.89] were more likely to have obesity than males. Respondents who had secondary [OR: 3.01; CI: 1.64, 5.53] or higher education [OR: 1.93; CI: 1.10, 3.36] in comparison with the respondents without any formal education were more likely to suffer from obesity. Surprisingly, the urban non-migrants who currently smoke [OR: 0.30; CI: 0.16, 0.54] had a lower likelihood of obesity than their counterpart who never consumed tobacco. Respondents with mobility impairment [OR: 1.79; CI: 1.15, 2.77] had a higher likelihood of obesity than their counterparts. Middle aged rural-to-urban migrants are likely to have higher obesity than their old-aged counterparts [OR: 2.73; CI: 1.84, 4.04]. Similar to the non-migrants, female rural-to-urban migrants were likely to have higher obesity than male migrants [OR: 2.08; CI: 1.50, 2.89]. Migrants with primary [OR: 1.55; CI: 1.01, 2.36] and secondary [OR: 2.05; CI: 1.23, 3.41] education are more likely to have obesity than their illiterate counterparts. In comparison with the migrants who were currently working, retired migrants were likely to have higher obesity [OR: 1.41; CI: 0.98, 2.03]. Similarly, migrants who were physically inactive were more likely to suffer from obesity than those who were physically active [OR: 1.26; CI: 0.97,1.65]. Migrants with mobility impairment [OR: 1.62; CI: 1.10, 2.39] were more likely to suffer from obesity than their counterparts who did not have such impairment.

Results from multivariable quantile regression presented in Table 4 revealed that, overall, in comparison with the rural non-migrants, rural-to-urban migrants had a higher risk of obesity. Logistic regression estimates present recent migrants (0–5 years) were 1.91 times [OR: 1.91; CI: 1.22, 2.99], 6–10 years of migrants were 2.05 times [OR: 2.05; CI: 1.15, 2.62] and more than 10 years of migrants in urban areas were 2.40 times [OR: 2.40; CI: 1.10, 3.15] more likely to have obesity than their rural non-migrant counterparts. Within the 10th quantile, in comparison with the rural-non-migrants, 1.18 times

**Table 1.** Sociodemographic characteristics of Rural non-migrant, Urban non-migrant, and rural-to-urban migrant population aged 45+ in India, LASI Wave 1 (2017−18).

| Background characteristics | Rural non-migrant | | Urban non-migrant | | Rural-urban migrant | |
|---|---|---|---|---|---|---|
| | Sample | Percentage | Sample | Percentage | Sample | Percentage |
| **Age** | | | | | | |
| Middle-aged (44–58) | 9012 | 51.11 | 3495 | 56.14 | 4245 | 54.15 |
| Elderly (59+) | 8619 | 48.89 | 2730 | 43.86 | 3594 | 45.85 |
| **Sex** | | | | | | |
| Male | 12786 | 72.49 | 3750 | 60.24 | 2778 | 35.44 |
| Female | 4846 | 27.51 | 2475 | 39.76 | 5061 | 64.56 |
| **Education** | | | | | | |
| Illiterate | 8721 | 49.46 | 1597 | 25.65 | 2761 | 35.23 |
| Primary | 3589 | 20.36 | 1085 | 17.43 | 1597 | 20.37 |
| Secondary | 4071 | 23.10 | 2216 | 35.6 | 2406 | 30.69 |
| Higher | 1250 | 7.09 | 1327 | 21.32 | 1075 | 13.71 |
| **Work** | | | | | | |
| Never worked | 2207 | 12.53 | 1509 | 24.24 | 3230 | 41.21 |
| Retired | 4502 | 25.53 | 1692 | 27.18 | 1979 | 25.24 |
| Currently Working | 10922 | 61.94 | 3024 | 48.58 | 2630 | 33.55 |
| **MPCE Quantile** | | | | | | |
| Poorest | 3673 | 20.83 | 1524 | 24.48 | 1653 | 21.09 |
| Poor | 3631 | 20.59 | 1332 | 21.40 | 1657 | 21.14 |
| Middle | 3578 | 20.29 | 1231 | 19.78 | 1577 | 20.12 |
| Richer | 3433 | 19.47 | 1161 | 18.65 | 1550 | 19.77 |
| Richest | 3316 | 18.81 | 977 | 15.69 | 1402 | 17.88 |
| **Caste** | | | | | | |
| SC/ST | 7578 | 42.98 | 1727 | 27.74 | 2217 | 28.28 |
| OBC | 6723 | 38.13 | 2651 | 42.59 | 3054 | 38.96 |
| Others | 3330 | 18.89 | 1847 | 29.67 | 2568 | 32.76 |
| **Religion** | | | | | | |
| Hindu | 12855 | 72.91 | 4227 | 67.9 | 5832 | 74.40 |
| Muslim | 1550 | 8.79 | 1111 | 17.85 | 1101 | 14.04 |
| Others | 3224 | 18.30 | 887 | 14.25 | 906 | 11.56 |
| **Physical activity** | | | | | | |
| Active | 11252 | 63.82 | 3862 | 62.04 | 4793 | 61.14 |
| Inactive | 6379 | 36.18 | 2363 | 37.96 | 3046 | 38.86 |
| **Alcohol consumption** | | | | | | |
| Never consumed | 12586 | 71.39 | 5139 | 82.55 | 6976 | 88.99 |
| Infrequent | 2851 | 16.17 | 620 | 9.96 | 529 | 6.75 |
| Frequent | 1953 | 11.08 | 411 | 6.60 | 295 | 3.76 |
| Heavy drinker | 241 | 1.37 | 55 | 0.88 | 39 | 0.50 |
| **Tobacco consumption** | | | | | | |
| Non-smoker | 13660 | 77.48 | 5418 | 87.04 | 7121 | 90.84 |
| Current smoker | 3971 | 22.52 | 807 | 12.96 | 718 | 9.16 |
| **Self-rated health** | | | | | | |
| Very good | 804 | 4.56 | 370 | 5.94 | 436 | 5.56 |
| Good | 7109 | 40.32 | 2454 | 39.42 | 3079 | 39.27 |
| Fair | 7158 | 40.59 | 2521 | 40.50 | 3057 | 39.01 |

*(Continued)*

| Background characteristics | Rural non-migrant | | Urban non-migrant | | Rural-urban migrant | |
|---|---|---|---|---|---|---|
| | Sample | Percentage | Sample | Percentage | Sample | Percentage |
| Poor | 2321 | 13.17 | 790 | 12.69 | 1139 | 14.53 |
| Very poor | 239 | 1.36 | 90 | 1.45 | 128 | 1.63 |
| **Difficulty in ADL** | | | | | | |
| No | 16638 | 94.37 | 5905 | 94.86 | 7278 | 92.84 |
| Yes | 993 | 5.63 | 320 | 5.14 | 561 | 7.16 |
| **Mobility problem** | | | | | | |
| No | 7887 | 44.73 | 3034 | 48.74 | 2934 | 37.42 |
| Yes | 9744 | 55.27 | 3191 | 51.26 | 4905 | 62.58 |
| **Sleep problem** | | | | | | |
| No | 15785 | 89.53 | 5689 | 91.39 | 6992 | 89.20 |
| Yes | 1846 | 10.47 | 536 | 8.61 | 847 | 10.80 |
| **Total** | **17631** | **55.63** | **6225** | **19.64** | **7839** | **24.73** |

**Note**: Counts are un-weighted, and percentages are weighted, MPCE: Monthly per capita consumption expenditure, SC/ST: Scheduled caste/scheduled tribe, OBC: Other backward classes, ADL: Activities of daily living.

higher BMI was observed in recent migrants, and it increased with the duration of residence in urban areas. The likelihood of obesity was found to be much higher in the 75th and 90th quantiles. In the 90th quantiles, recent migrants (≤5 years) were 1.94 times, 6–10 years migrants were 1.73 times and Migrants who were living ≥10 in urban areas were 2.32 times more likely to have elevated BMI than their rural non-migrant counterparts. For more information for other covariates please refer to Fig 3.

## Discussion

The current study investigated the effect of rural-to-urban migration and migration duration in urban areas on obesity among middle-aged and older Indian adults. The findings revealed that migrants had higher BMI values and a higher risk of obesity than rural non-migrants. This study backs up the hypothesis that a longer period of urban residency among rural-to-urban migrants is associated with an increased risk of obesity. Even after adjusting for potential confounding factors, this association remained significant.

Prior research has consistently established the association between the length of urban residency and the prevalence of obesity among the rural-to-urban migrant population. This trend has been observed not only among international migrants, as demonstrated by studies conducted in the United States [30,53] but also among rural-to-urban migrants in many low- and middle-income countries [4,29]. Individuals who had lived in the United States for more than 15 years, for example, had a threefold increase in the risk of being overweight (BMI 25 kg/m$^2$) compared to those who had lived in the country for 10 years [45]. In Peru, for every 10-year increment in urban residence duration, individuals migrating from rural areas showed an average 12% increase in the prevalence of obesity [54].

The obesogenic nature of urban environments might account for the increased risk of obesity among individuals residing in urban environments, including both urban residents and rural-to-urban migrants. Multiple cross-sectional and longitudinal studies suggest that both migrants and urban dwellers have higher levels of physical inactivity and sedentary behaviour than their rural counterparts [26,55]. In our study, we found that migrants had the highest prevalence of low physical activity, followed by urban and rural locals. Obesity is frequently associated with an imbalance between physical activity and diet, prompting the consideration of unhealthy diets as an alternative explanation. Unfortunately, our study

**Table 2. Prevalence of obesity stratified by migration status among the middle aged and older adults (aged 45+) in India, LASI Wave 1, (2017−18).**

| Background characteristics | Rural non-migrant | Urban non-migrant | Rural-urban Migrant |
|---|---|---|---|
| **Age** | | | |
| Middle-aged (44–58)* | 2.82 | 13.85 | 15.70 |
| Elderly (59+)* | 2.39 | 12.38 | 9.48 |
| **Sex** | | | |
| Male* | 2.00 | 7.04 | 6.11 |
| Female* | 4.74 | 23.98 | 15.50 |
| **Education** | | | |
| Illiterate* | 1.83 | 8.90 | 9.87 |
| Primary* | 2.08 | 9.45 | 12.55 |
| Secondary* | 3.70 | 19.36 | 16.43 |
| Higher* | 5.40 | 10.80 | 12.86 |
| **Work** | | | |
| Never worked* | 5.19 | 29.34 | 16.98 |
| Retired* | 3.10 | 7.90 | 10.99 |
| Currently Working* | 2.02 | 8.57 | 8.27 |
| **MPCE Quantile** | | | |
| Poorest* | 1.03 | 10.70 | 7.89 |
| Poor* | 1.56 | 9.17 | 14.31 |
| Middle* | 2.30 | 10.38 | 11.94 |
| Richer* | 3.84 | 14.23 | 10.95 |
| Richest* | 4.88 | 23.97 | 20.64 |
| **Caste** | | | |
| SC/ST* | 1.61 | 7.88 | 10.03 |
| OBC* | 2.43 | 14.66 | 12.08 |
| Others | 4.51 | 13.51 | 15.31 |
| **Religion** | | | |
| Hindu* | 2.50 | 12.11 | 12.16 |
| Muslim* | 2.25 | 17.25 | 18.56 |
| Others* | 4.39 | 15.91 | 10.06 |
| **Physical activity** | | | |
| Active* | 2.39 | 14.18 | 12.84 |
| Inactive* | 2.98 | 11.35 | 12.41 |
| **Alcohol consumption** | | | |
| Never consumed* | 2.95 | 14.70 | 13.32 |
| Infrequent* | 2.03 | 4.17 | 7.42 |
| Frequent* | 1.03 | 4.29 | 3.75 |
| Heavy drinker* | 0.22 | 5.62 | 3.29 |
| **Tobacco consumption** | | | |
| Non-smoker* | 3.16 | 14.71 | 13.44 |
| Current smoker | 0.81 | 2.35 | 2.55 |
| **Self-rated health** | | | |
| Very good* | 2.89 | 15.47 | 15.89 |
| Good* | 2.37 | 9.04 | 13.24 |
| Fair* | 2.51 | 18.26 | 11.27 |

*(Continued)*

**Table 2.** (Continued)

| Background characteristics | Rural non-migrant | Urban non-migrant | Rural-urban Migrant |
|---|---|---|---|
| Poor* | 2.93 | 8.21 | 13.26 |
| Very poor* | 5.74 | 8.80 | 20.87 |
| **Difficulty in ADL** | | | |
| No* | 2.44 | 13.18 | 12.55 |
| Yes* | 4.66 | 13.40 | 13.91 |
| **Mobility problem** | | | |
| No* | 1.89 | 8.88 | 9.60 |
| Yes* | 3.11 | 16.81 | 14.36 |
| **Sleep problem** | | | |
| No* | 2.56 | 13.37 | 12.63 |
| Yes* | 2.87 | 11.18 | 13.08 |
| **Total*** | **2.59** | **13.19** | **12.66** |

**Note:** estimates are weighted; MPCE: Monthly per capita consumption expenditure, SC/ST: Scheduled caste/scheduled tribe, OBC: Other backward classes, ADL: Activities of daily living. * If p-value of chi2 test <0.05.

lacks data on dietary patterns. Nonetheless, existing research indicates that urban people consume more saturated fat and energy-dense foods than migrants and rural people [55].

Another possible explanation for the findings is that the combination of socio-economic status (SES) and exposure to a western lifestyle influences weight. Previous studies found the link between socio-economic status and obesity varies based on a country's level of development. In less developed countries, people belonging to higher Socioeconomic status are often associated with a higher likelihood of obesity. In more developed countries, it is the opposite, where higher socio-economic status (SES) is frequently associated with a lower likelihood of obesity [27,30]. The socioeconomic disparities between rural and urban areas in India significantly impact the prevalence of obesity, particularly among rural-to-urban migrants, as demonstrated by the results presented in this study. In urban environments, migrants are often introduced to greater economic opportunities, which may increase their per capita consumption expenditure [54]. This rise in economic status can paradoxically contribute to higher obesity rates. Migrants transitioning from labour-intensive rural jobs to more sedentary urban professions face a decrease in daily physical activity, while simultaneously gaining access to a wider array of calorie-dense, processed foods that are more readily available in urban settings [56]. Migrants within higher wealth quantiles exhibit increased risks of obesity compared to their peers in lower quantiles. This pattern suggests that as economic status improves, so does the propensity towards adopting urban lifestyles that promote higher caloric intake and reduced physical activity [57]. Such shifts in lifestyle are significant contributors to the obesity disparity observed between individuals from different economic backgrounds, especially as they adapt to the new urban socioeconomic environment [4,55]. In terms of the monthly per capita consumption expenditure, more affluent individuals are more susceptible to obesity than their less affluent counterparts.

These risk assessments are unaffected by levels of physical activity, implying that dietary habits may be an important factor [58,59]. As a result, there may be a need to improve access to or awareness of the importance of healthy diets, particularly among those with higher socio-economic status. However, we observed a higher risk of obesity in middle-aged migrants compared to those in old age. One possible explanation is that, compared to migrants who migrate in old age, those who move during middle age tend to encounter and adopt more aspects of the urban lifestyle after migration. "This is due to their better chance of adapting to the urban lifestyle and practices. On the other hand, older migrants, who usually come to the urban areas to reunite with family, are less likely to adapt to the unfamiliar urban culture." [29].

**Table 3. Logistic regression estimates for the determinants of obesity for Rural non-migrant, urban non-migrant, and rural-to-urban migrant population aged 45+ in India, LASI Wave 1 (2017−18).**

| Background characteristics | Rural non-migrant | | Urban non-migrant | | Rural-to-urban migrant | |
|---|---|---|---|---|---|---|
| | OR | 95% CI | OR | 95% CI | OR | 95% CI |
| **Age** | | | | | | |
| Old-aged | Ref | | Ref | | Ref | |
| Middle-aged | 1.31 | (0.99, 1.74) | 1.46 | (0.85, 2.49) | 2.08*** | (1.50, 2.89) |
| **Sex** | | | | | | |
| Male | Ref | | Ref | | Ref | |
| Female | 2.86*** | (2.07, 3.96) | 2.33*** | (1.52, 3.58) | 2.73*** | (1.84, 4.04) |
| **Education** | | | | | | |
| Illiterate | Ref | | Ref | | Ref | |
| Primary | 1.40 | (0.97, 2.02) | 1.46 | (0.9, 2.38) | 1.55* | (1.01, 2.36) |
| Secondary | 2.53*** | (1.8, 3.56) | 3.01*** | (1.64, 5.53) | 2.05* | (1.23, 3.41) |
| Higher | 3.45*** | (2.23, 5.36) | 1.93** | (1.1, 3.36) | 1.73 | (0.95, 3.18) |
| **Work** | | | | | | |
| Currently Working | Ref | | Ref | | Ref | |
| Never worked | 1.27 | (0.84, 1.93) | 1.76* | (1.04, 2.98) | 1.40 | (0.91, 2.14) |
| Retired | 1.36 | (0.97, 1.89) | 0.87 | (0.53, 1.41) | 1.41* | (0.98, 2.03) |
| **MPCE quantile** | | | | | | |
| Richest | Ref | | Ref | | Ref | |
| Poorest | 1.42 | (0.86, 2.34) | 0.75 | (0.41, 1.36) | 1.75*** | (1.21, 2.54) |
| Poor | 1.93* | (1.2, 3.09) | 0.78 | (0.41, 1.49) | 1.45* | (1.00, 2.09) |
| Middle | 3.15** | (1.97, 5.04) | 1.19 | (0.61, 2.31) | 1.23 | (0.78, 1.93) |
| Richer | 3.16*** | (2.03, 4.91) | 1.96 | (0.79, 4.85) | 2.62*** | (1.47, 4.68) |
| **Caste** | | | | | | |
| Others | Ref | | Ref | | Ref | |
| SC/ST | 0.49*** | (0.35, 0.71) | 0.79 | (0.48, 1.31) | 0.79 | (0.54, 1.15) |
| OBC | 0.68** | (0.51, 0.91) | 1.07 | (0.67, 1.72) | 0.79 | (0.58, 1.09) |
| **Religion** | | | | | | |
| Others | Ref | | Ref | | Ref | |
| Hindu | 0.75 | (0.51, 1.1) | 1.90** | (1.08, 3.34) | 1.61 | (0.97, 2.67) |
| Muslim | 1.78*** | (1.2, 2.63) | 1.59 | (0.94, 2.68) | 0.73 | (0.41, 1.29) |
| **Physical activity** | | | | | | |
| Active | Ref | | Ref | | Ref | |
| Inactive | 1.12 | (0.84, 1.5) | 1.07 | (0.71, 1.62) | 1.26* | (0.97, 1.65) |
| **Alcohol consumption** | | | | | | |
| Never consumed | Ref | | Ref | | Ref | |
| Infrequent | 1.13 | (0.76, 1.68) | 0.58 | (0.32, 1.05) | 1.36 | (0.73, 2.55) |
| Frequent | 0.66 | (0.40, 1.10) | 0.66 | (0.29, 1.53) | 0.81 | (0.37, 1.78) |
| heavy drinker | 0.14** | (0.04, 0.57) | 1.21 | (0.33, 4.49) | 0.76 | (0.12, 4.73) |
| **Tobacco consumption** | | | | | | |
| Non-smoker | Ref | | Ref | | Ref | |
| Current smoker | 0.39*** | (0.25, 0.61) | 0.30*** | (0.16, 0.54) | 0.36*** | (0.19, 0.68) |
| **Self-rated health** | | | | | | |
| Very poor | Ref | | Ref | | Ref | |
| Very good | 0.70 | (0.25, 1.97) | 1.87 | (0.52, 6.68) | 1.04 | (0.47, 2.27) |

*(Continued)*

**Table 3.** (Continued)

| Background characteristics | Rural non-migrant | | Urban non-migrant | | Rural-to-urban migrant | |
|---|---|---|---|---|---|---|
| | OR | 95% CI | OR | 95% CI | OR | 95% CI |
| Good | 0.69 | (0.29, 1.64) | 0.94 | (0.28, 3.15) | 0.69 | (0.35, 1.37) |
| Fair | 0.64 | (0.27, 1.53) | 1.84 | (0.57, 5.95) | 0.57 | (0.29, 1.12) |
| Poor | 0.69 | (0.29, 1.67) | 0.88 | (0.27, 2.83) | 0.65 | (0.33, 1.29) |
| **Difficulty in ADL** | | | | | | |
| No | Ref | | Ref | | Ref | |
| Yes | 1.57*** | (1.01, 2.43) | 0.85 | (0.43, 1.69) | 1.01 | (0.63, 1.59) |
| **Mobility problem** | | | | | | |
| No | Ref | | Ref | | Ref | |
| Yes | 1.52** | (1.16, 1.99) | 1.79** | (1.15, 2.77) | 1.62* | (1.10, 2.39) |
| **Sleep problem** | | | | | | |
| No | Ref | | Ref | | Ref | |
| Yes | 0.97 | (0.67, 1.42) | 0.91 | (0.54, 1.52) | 1.02 | (0.72, 1.37) |

**Note:** Ref Reference, OR Odds ratio, CI Confidence Interval, SC/ST Schedule caste/ Schedule tribe, OBC Other backward caste, MPCE Monthly per capita consumption expenditure, *** p ≤ 0.001, ** p < 0.01 and * p ≤ 0.05.

**Table 4. Estimates of logistic and quantile regression for the risk of obesity by duration of residence in urban areas among the migrants aged 45+ in India, LASI wave 1 (2017−18).**

| Duration of urban residence | OR (95% CI) | 10th percentile (95% CI) | 25th percentile (95% CI) | 50th percentile (95% CI) | 75th percentile (95% CI) | 90th percentile (95% CI) |
|---|---|---|---|---|---|---|
| Rural non-migrant | Ref. | Ref. | Ref. | Ref. | Ref. | Ref. |
| Rural-urban migrant (Up to 5 years) | 1.91** (1.22, 2.99) | 1.18*** (0.74, 1.65) | 1.78*** (1.29, 2.26) | 1.98*** (1.44, 2.51) | 1.93*** (1.72, 2.70) | 1.94*** (1.34, 2.52) |
| Rural-urban migrant (6–10 years) | 2.05* (1.15, 3.62) | 1.69*** (1.24, 2.13) | 1.76*** (1.39, 2.13) | 1.95*** (1.49, 2.41) | 2.18*** (1.50, 2.86) | 1.73*** (1.09, 2.32) |
| Rural-urban migrant (more than 10 years) | 2.40*** (1.84, 3.15) | 1.50*** (1.31, 1.69) | 1.92*** (1.77, 2.07) | 2.20*** (2.03, 2.38) | 2.21*** (1.97, 2.46) | 2.32*** (1.99, 2.65) |
| Urban non-migrant | 3.31*** (2.39, 4.58) | 1.31*** (1.12, 1.50) | 1.69*** (1.53, 1.86) | 1.95*** (1.80, 2.11) | 2.24*** (2.12, 2.36) | 2.24*** (2.02, 2.47) |

**Note**- Ref: Reference; OR: Odds ratio; CI: Confidence interval; Estimates are weighted and adjusted for all selected covariates; P-values are presented as ***<0.001, **<0.01, *<0.05.

There was no uniform increased risk of developing obesity at the highest level of educational attainment, similar results were found in the United States [58,59]. The trends in obesity among rural-to-urban migrants, particularly when viewed through the lens of education, reveal a complex, nuanced story. As these migrants gain education, moving from illiteracy to primary and then secondary levels, we see an initial rise in obesity rates. This increase likely reflects an improvement in their economic situation, which often brings with it the symbols of urban life, including easier access to calorie-dense foods and perhaps less physically demanding jobs. However, education is not just about earning potential; it also involves learning about the world and making informed decisions [60]. By the time migrants reach higher educational levels, this increased awareness starts to show its effects. Interestingly, at the higher education level, the trend in obesity begins to reverse slightly. This suggests that with higher education comes a greater understanding of nutrition and health, as well

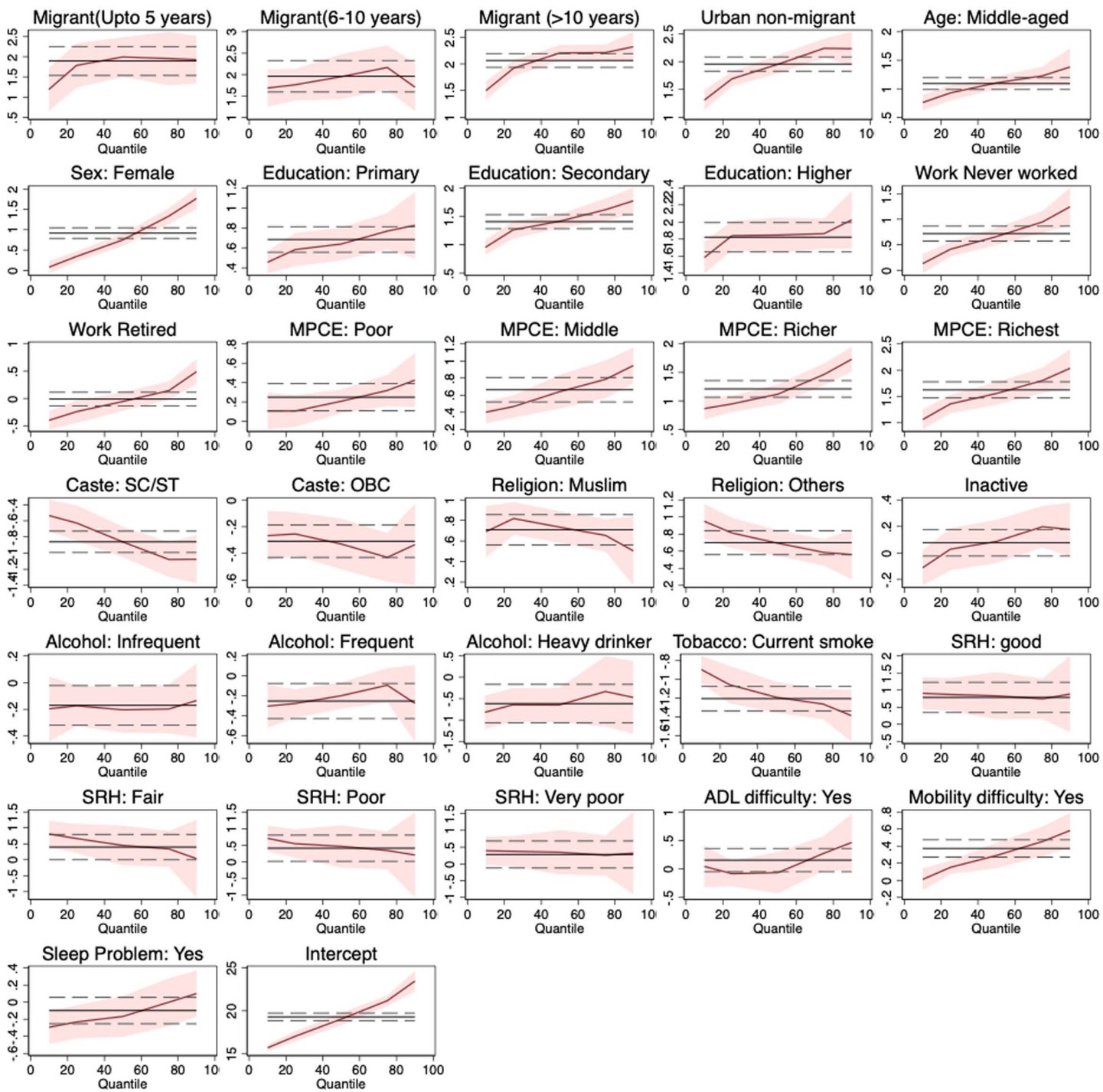

**Fig 3. Multivariable quantile regression predicting obesity at the 0.10–0.90 quantile. Note:** Obesity estimates from quantile regression are represented by the solid red line and their 95% confidence intervals are shown by the shaded red region. The solid black line is the effect size estimate from conventional linear regression analysis with its 95% confidence intervals shown by the black dashed lines.

as better access to health-promoting resources like physical activity and healthier food options. These individuals are likely to be more integrated into networks that value and support healthier lifestyle choices, which can have a significant impact on behaviour [61]. Given the potential for education to be a focal point for preventive measures, more research into the relationship between educational level and the risk of developing obesity is warranted. We found that older adults with disabilities are more susceptible to obesity. Existing literature identifies reduced mobility as a key contributor, limiting physical activity and predisposing older individuals with disabilities to weight gain [62]. Additionally, psychosocial factors, including heightened stress and depression, which often accompany a disability, increase the risk of obesity [7]. Our findings reveal notable sex differences in the relationship between migration from rural to urban areas and the risk of obesity. Female migrants have a higher risk of obesity than male migrants in any duration of urban residence (See Table 7). This finding adds to existing research indicating a higher prevalence of obesity in women migrants in the Indian scenario [4]. It demonstrates that women are significantly more vulnerable than men to the negative effects of rural-to-urban migration on obesity status. Several contributing factors include social and cultural norms that may limit physical activity among women, dietary changes that impact women differently due to their distinct nutritional needs, and the stress associated with relocation and new social roles [63]. Moreover, biological factors such as a genetic predisposition among South Asian women to store more visceral fat, higher insulin resistance, and variations in leptin sensitivity further exacerbate this risk, making the transition to urban life a particularly challenging environment for managing healthy body weight [64,65].

Our study has a few limitations that need recognition. The use of a cross-sectional design restricts our ability to track changes in obesity over time in association with migration. Another significant limitation is the lack of information on dietary practices in our study, hampering our ability to fully grasp the connection between dietary habits and obesity in the population we examined. Despite the limitations, our study has strengths. It includes a nationally representative large sample. The study features a robust methodology with multiple measurements for the research objective. Unlike BMI, which cannot differentiate body fat deposition, our study incorporated waist-to-hip ratio and waist circumference. These additions allowed the evaluation of visceral fat deposition in the abdomen, a significant risk factor for several diseases. Furthermore, the use of quantile regression allowed for a more in-depth investigation of the relationship between migration and BMI across the full BMI distribution. We carefully compared rural-to-urban migrants with their rural non-migrant counterparts to ensure baseline similarity in their characteristics, a deliberate approach lacking in the comparison to urban natives in order to limit any selection bias.

## Sensitivity analysis

We incorporated a more responsive measure of obesity by including abdominal obesity (Table 5). The importance of body fat distribution as a critical risk factor for obesity-related complications has been highlighted by earlier empirical studies [51]. Waist circumference and waist-to-hip ratio provide additional and independent information to BMI, enhancing its predictive ability for morbidity and risk of death [66,67].

The effect of rural-to-urban migration on the people with morbid obesity (BMI > 40 kg/m2) has also been studied (Table 6). Since, the global definition of obesity (BMI ≥ 30 kg/m2) is primarily derived from data pertaining to Caucasian populations [68]. We investigated the independent and adjusted impact of rural-to-urban migration on obesity using the suggested BMI cutoff (BMI ≥ 25 kg/m$^2$) for the Asian population [69] (Table 6). Nonetheless, Asian populations, particularly South Asians, show higher visceral adipose tissue at a given BMI. We provided information on individuals with no obesity, who were underweight (BMI < 18.5 kg/m$^2$) based on the migration status and place of residence (Figure S3 in S1 File), which enabled us to gain a more comprehensive view of the distribution of body weight among migrant and non-migrant population.

We additionally examined the effect of migration on obesity by stratifying the analysis by sex to explore potential sex differences in obesity risk patterns (Table 7). The findings revealed a consistently stronger association between rural-to-urban migration and obesity across both sexes, with notable differences in the magnitude of risk. For instance, women

**Table 5. Estimates of Logistic and Quantile regression for the people with risk of abdominal obesity by duration of residence in urban areas among the middle-aged and older adults (aged 45+) migrants in India, LASI wave 1 (2017−18).**

| Duration of migration | OR (95% CI) | 10th percentile (95% CI) | 25th percentile (95% CI) | 50th percentile (95% CI) | 75th percentile (95% CI) | 90th percentile (95% CI) |
|---|---|---|---|---|---|---|
| **Panel A- Abdominal obesity (High waist circumference)** | | | | | | |
| Rural non-migrant | Ref. | Ref. | Ref. | Ref. | Ref. | Ref. |
| Rural-urban migrant (Up to 5 years) | 3.38***, (1.95, 5.85) | 4.20*** (2.47, 5.92) | 5.61*** (4.41, 6.82) | 5.73*** (4.48, 6.98) | 5.14*** (3.05, 7.23) | 5.68*** (3.98, 7.39) |
| Rural-urban migrant (6–10 years) | 2.86***, (1.86, 4.40) | 5.89*** (3.98, 7.79) | 5.57*** (4.40, 6.74) | 5.59*** (3.59, 7.59) | 5.70*** (4.80, 5.59) | 5.84*** (4.16, 7.50) |
| Rural-urban migrant (more than 10 years) | 2.57***, (2.19, 3.02) | 5.05*** (4.47, 5.62) | 6.33*** (5.79, 6.87) | 6.77*** (6.35, 7.18) | 6.14*** (5.53, 6.75) | 6.32*** (5.65, 6.99) |
| Urban non-migrant | 2.57***, (2.14, 3.09) | 4.58*** (4.14, 5.100) | 5.38*** (4.86, 5.90) | 5.58*** (5.12, 6.04) | 5.35*** (4.80, 5.90) | 5.57*** (4.94, 6.21) |
| **Panel B- Abdominal obesity (High waist-to-hip ratio)** | | | | | | |
| Rural non-migrant | Ref. | Ref. | Ref. | Ref. | Ref. | Ref. |
| Rural-urban migrant (Up to 5 years) | 1.41*** (1.10, 1.80) | 1.54** (0.19, 2.90) | 1.44*** (0.95, 1.92) | 1.26*** (0.68, 1.83) | 1.05*** (0.32, 1.78) | 1.09*** (0.06, 2.12) |
| Rural-urban migrant (6–10 years) | 1.54*** (1.19, 1.99) | 1.79*** (0.82, 2.75) | 1.83*** (1.12, 2.53) | 1.66*** (0.96, 2.35) | 1.98*** (1.45, 2.50) | 1.18*** (0.28, 2.08) |
| Rural-urban migrant (more than 10 years) | 1.86*** (1.71, 2.04) | 2.61*** (2.17, 3.06) | 2.51*** (2.20, 2.82) | 2.15*** (1.88, 2.43) | 1.96*** (1.66, 2.26) | 1.69*** (1.28, 2.11) |
| Urban non-migrant | 1.53*** (1.40, 1.66) | 1.93*** (1.64, 2.23) | 1.88*** (1.62, 2.15) | 1.71*** (1.48, 1.94) | 1.66*** (1.43, 1.89) | 1.63*** (1.25, 2.02) |

**Note**- Ref: Reference; OR: Odds ratio; CI: Confidence interval; Estimates are weighted and adjusted for all selected covariates; P-values are presented as ***<0.001, **<0.01, *<0.05.

**Table 6. Estimates of Logistic regression for the people with risk of obesity (BMI ≥ 25 kg/m²) and people with morbid obesity (BMI ≥ 40 kg/m²) by duration of residence in urban areas among the migrants aged 45+ in India, LASI wave 1 (2017−18).**

| Duration of migration | People with obesity (BMI ≥ 25 kg/m²)[*] | | People with morbid obesity (BMI ≥ 40 kg/m²) | |
|---|---|---|---|---|
| | UOR (95% CI) | AOR (95% CI) | UOR (95% CI) | AOR (95% CI) |
| Rural non-migrant | Ref. | Ref. | Ref. | Ref. |
| Rural-urban migrant (Up to 5 years) | 3.97*** (2.49, 6.32) | 2.49*** (1.47, 4.22) | 7.99* (1.32, 48.17) | 3.75* (0.57, 24.67) |
| Rural-urban migrant (6–10 years) | 4.60*** (2.86, 7.41) | 2.63*** (1.68, 4.42) | 2.56 (0.36, 18.01) | 1.23 (0.17, 8.79) |
| Rural-urban migrant (more than 10 years) | 4.08*** (3.43, 4.85) | 2.52*** (2.16, 2.95) | 8.38 (3.61, 19.46) | 3.64 (1.39, 9.53) |
| Urban non-migrant | 4.01*** (3.29, 4.89) | 2.83*** (2.39, 3.35) | 4.91*** (1.99, 12.10) | 2.88* (1.09, 7.59) |

**Note**- UOR: Unadjusted Odds Ratio; AOR: Adjusted Odds Ratio; *: Based on obesity cutoff for Asians; CI: Confidence interval; Ref: Reference; P-values are presented as ***<0.001, **<0.01, *<0.05.

**Table 7. Estimates from Logistic and Quantile Regression by Sex of Respondents on the Risk of Obesity by Duration of Migration Among Middle-Aged and Older Adults in India, LASI Wave 1 (2017–18).**

| Duration of migration | OR | 10th percentile | 25th percentile | 50th percentile | 75th percentile | 90th percentile |
|---|---|---|---|---|---|---|
| | (95% CI) | (95% CI) | (95% CI) | (95% CI) | (95% CI) | (95% CI) |
| **Panel A – Male** | | | | | | |
| Rural non-migrant | Ref. | Ref. | Ref. | Ref. | Ref. | Ref. |
| Rural-urban migrant (Up to 5 years) | 1.26*** (0.43, 3.68) | 1.36*** (0.19, 1.87) | 1.40*** (0.67, 2.14) | 1.46*** (0.50, 2.83) | 1.55*** (0.18, 2.53) | 1.56*** (0.55, 2.17) |
| Rural-urban migrant (6–10 years) | 2.23*** (1.09, 4.56) | 1.40*** (1.30, 2.31) | 1.59*** (1.06, 2.13) | 1.82*** (0.92, 2.71) | 1.89*** (1.55, 2.23) | 1.97*** (1.04, 2.50) |
| Rural-urban migrant (more than 10 years) | 2.27*** (1.64, 3.15) | 1.42*** (1.23, 1.62) | 1.88*** (1.69, 2.08) | 2.11*** (1.84, 2.38) | 2.19*** (1.93, 2.45) | 2.19*** (1.78, 2.61) |
| Urban non-migrant | 2.42*** (1.73, 3.38) | 1.49*** (0.83, 1.86) | 1.92*** (1.32, 2.72) | 2.17*** (1.45, 2.88) | 2.23*** (1.77, 3.20) | 2.28*** (1.69, 2.98) |
| **Panel B- Female** | | | | | | |
| Rural non-migrant | Ref. | Ref. | Ref. | Ref. | Ref. | Ref. |
| Rural-urban migrant (Up to 5 years) | 2.27*** (1.36, 3.81) | 1.50*** (0.91, 2.08) | 1.56*** (1.03, 2.68) | 1.98*** (1.29, 2.97) | 2.19*** (1.23, 3.15) | 2.16*** (1.77, 2.59) |
| Rural-urban migrant (6–10 years) | 2.06*** (1.02, 4.33) | 1.55*** (0.99, 2.12) | 1.74*** (1.07, 2.41) | 2.08*** (1.44, 2.72) | 2.43*** (1.76, 4.10) | 2.17*** (0.99, 2.96) |
| Rural-urban migrant (more than 10 years) | 2.65*** (1.89, 3.70) | 1.61*** (1.27, 1.84) | 1.93*** (1.69, 2.17) | 2.27*** (2.03, 2.50) | 2.48*** (2.00, 2.97) | 2.40*** (2.05, 2.75) |
| Urban non-migrant | 3.96*** (2.50, 6.26) | 1.83*** (1.53, 2.14) | 2.06*** (1.83, 2.30) | 2.49*** (2.18, 2.79) | 2.86*** (2.52, 3.19) | 2.86*** (2.26, 3.46) |

**Note**- Ref: Reference; OR: Odds ratio; CI: Confidence interval; Estimates are weighted and adjusted for all selected covariates; P-values are presented as ***<0.001, **<0.01, *<0.05.

residing in urban areas had significantly elevated odds of obesity. Although male migrants also showed rising obesity risks with increased urban exposure the magnitude was slightly lower compared to females. Both sexes exhibited a clear gradient, with obesity risk increasing as the duration of urban residence lengthened.

Similarly, stratified analysis by age groups (44–59+) confirmed that both age cohorts experienced an increased risk of obesity with longer urban exposure (Table 8). However, the effect was more pronounced among middle-aged adults, consistent with earlier findings.

## Policy recommendations

Given the rising rates of obesity in India, particularly among vulnerable groups like rural-to-urban migrants, there is a clear need for an exclusive and comprehensive health policy related to obesity. Migrants face significantly higher risks of obesity due to factors like prolonged urban exposure, unhealthy lifestyles, poor access to nutritious food, and socio-economic vulnerabilities. Middle-aged migrants, women, and those with low physical activity levels are particularly at risk. While existing government programs address obesity indirectly through broader non-communicable disease (NCD) strategies, there is no specific policy dedicated to tackling obesity, especially in the migrant population. The National Programme for Prevention and Control of Non-Communicable Diseases (NP-NCD) includes obesity as a modifiable risk factor but lacks focused interventions for migrants. To address this gap, the government should integrate migrant-specific obesity prevention and management strategies into the NP-NCD, including specific interventions like community-based education, health check-ups, and lifestyle modification programs. These efforts should be operationalized through Ayushman Bharat Health and Wellness Centres, which already provide NCD screening and management services. Targeted obesity screenings and

**Table 8. Estimates from Logistic and Quantile Regression by Age Group on the Risk of Obesity by Duration of Migration in India, LASI Wave 1 (2017–18).**

| Duration of migration | OR | 10th percentile | 25th percentile | 50th percentile | 75th percentile | 90th percentile |
|---|---|---|---|---|---|---|
| | (95% CI) | (95% CI) | (95% CI) | (95% CI) | (95% CI) | (95% CI) |
| **Panel A- Age 45–59** | | | | | | |
| Rural non-migrant | Ref. | Ref. | Ref. | Ref. | Ref. | Ref. |
| Rural-urban migrant (Up to 5 years) | 2.27*** (1.36, 3.81) | 1.50*** (0.91, 2.08) | 1.56*** (1.03, 2.68) | 1.98*** (1.29, 2.97) | 2.19*** (1.23, 3.15) | 2.16*** (1.77, 2.59) |
| Rural-urban migrant (6–10 years) | 2.06*** (1.02, 4.33) | 1.55*** (0.99, 2.12) | 1.74*** (1.07, 2.41) | 2.08*** (1.44, 2.72) | 2.43*** (1.76, 4.10) | 2.17*** (0.99, 2.96) |
| Rural-urban migrant (more than 10 years) | 2.65*** (1.89, 3.70) | 1.61*** (1.27, 1.84) | 1.93*** (1.69, 2.17) | 2.27*** (2.03, 2.50) | 2.48*** (2.00, 2.97) | 2.40*** (2.05, 2.75) |
| Urban non-migrant | 3.96*** (2.50, 6.26) | 1.83*** (1.53, 2.14) | 2.06*** (1.83, 2.30) | 2.49*** (2.18, 2.79) | 2.86*** (2.52, 3.19) | 2.86*** (2.26, 3.46) |
| **Panel B- Age 60+** | | | | | | |
| Rural non-migrant | Ref. | Ref. | Ref. | Ref. | Ref. | Ref. |
| Rural-urban migrant (Up to 5 years) | 2.32*** (1.33, 4.05) | 1.17*** (0.70, 1.64) | 1.43*** (0.93, 2.32) | 2.02*** (1.47, 2.57) | 2.10*** (1.31, 2.71) | 2.16*** (1.03, 3.30) |
| Rural-urban migrant (6–10 years) | 2.89*** (1.38, 5.38) | 1.23*** (1.03, 1.66) | 1.70*** (1.53, 1.91) | 2.06*** (1.53, 2.58) | 2.17*** (1.59, 2.76) | 2.31*** (1.88, 2.95) |
| Rural-urban migrant (more than 10 years) | 3.19*** (2.28, 4.47) | 1.45*** (1.20, 1.69) | 1.89*** (1.72, 2.02) | 2.11*** (1.84, 2.31) | 2.20*** (1.94, 2.42) | 2.46*** (2.03, 2.88) |
| Urban non-migrant | 3.85*** (2.80, 5.30) | 1.47*** (1.27, 1.66) | 1.91*** (1.58, 2.23) | 2.18*** (1.81, 2.55) | 2.38*** (2.05, 2.70) | 2.66*** (2.26, 3.07) |

**Note**- Ref: Reference; OR: Odds ratio; CI: Confidence interval; Estimates are weighted and adjusted for all selected covariates; P-values are presented as ***<0.001, **<0.01, *<0.05.

culturally sensitive health education, particularly for high-risk migrant groups, will ensure more effective intervention. Moreover, expanding existing national initiatives like the Fit India Movement and Eat Right India Campaign to directly address barriers faced by migrants is crucial. These programs can be adapted to promote physical activity and healthy eating in migrant communities, especially in urban slums where access to safe recreational spaces and affordable, healthy food is limited. Furthermore, the One Nation, One Ration Card scheme should be leveraged to ensure that migrants have access to basic nutritious food, reducing their reliance on calorie-dense, unhealthy diets that contribute to obesity. An exclusive obesity policy, designed with a focus on vulnerable populations like migrants, will allow India to more effectively curb the obesity epidemic.

## Conclusion

This study examined the impact of rural-to-urban migration on obesity among middle-aged and older adults in India and assessed whether the duration of urban residence influences obesity risk. The findings clearly indicate that rural-to-urban migrants are at a significantly higher risk of obesity compared to their rural non-migrant counterparts. Moreover, the risk increases progressively with longer durations of urban residence, underscoring a cumulative effect of urban exposure. These results emphasize the role of urban lifestyle transitions, characterized by physical inactivity and socio-economic changes, as key contributors to rising obesity levels among migrant populations. The study further highlights how obesity risks vary not just by migration status, but also across the BMI distribution. Given India's ongoing urban expansion, these findings point to an urgent need for specific public health interventions targeting rural-to-urban migrants, particularly those with prolonged urban exposure. Programs must account for age, socioeconomic background, and urban living conditions

to effectively mitigate obesity risks in this vulnerable demographic. Aligning obesity prevention strategies with the unique challenges of migrant adaptation can strengthen India's response to the growing burden of non-communicable diseases.

## Supporting information

**S1 Appendix.  Table A1 Multicollinearity test statistics.**
(DOCX)

**S1 File.**   Figure S1 Kernel destiny distribution of people with waist circumference score stratified by Migration status among the middle-aged and older adults (aged 45+) in India, LASI wave 1 (2017−18). **Figure S2** Sex stratified weighted prevalence of people with abdominal obesity by residence status (non-migrants) and duration of residence in urban areas among the middle-aged and older adults (aged 45+) migrants in India, LASI Wave 1 (2017−18). **Figure S3** Sex stratified weighted prevalence of people with underweight by migration status among the middle-aged and older adults (aged 45+) in India, LASI Wave 1 (2017−18). **Figure S4** Sex stratified weighted prevalence of people with high waist-to-hip ratio by migration status among the middle aged and older adults (aged 45+) in India, LASI Wave 1 (2017−18). **Figure S5** Selection criteria of the study sample.
(ZIP)

## Acknowledgments

We extend our gratitude to the International Institute for Population Science for providing the dataset.

## Author contributions

**Conceptualization:** Bittu Mandal, Kalandi Charan Pradhan.

**Data curation:** Bittu Mandal.

**Formal analysis:** Bittu Mandal.

**Investigation:** Bittu Mandal, Kalandi Charan Pradhan.

**Methodology:** Bittu Mandal, Kalandi Charan Pradhan.

**Resources:** Kalandi Charan Pradhan.

**Software:** Bittu Mandal.

**Supervision:** Kalandi Charan Pradhan.

**Validation:** Kalandi Charan Pradhan.

**Visualization:** Bittu Mandal.

**Writing – original draft:** Bittu Mandal.

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
