## [Decision Letter · Decision Letter 0]

Dear Dr. Mandal,

Thank you for submitting your manuscript to PLOS ONE. After careful consideration, we feel that it has merit but does not fully meet PLOS ONE’s publication criteria as it currently stands. Therefore, we invite you to submit a revised version of the manuscript that addresses the points raised during the review process.

We look forward to receiving your revised manuscript.

Kind regards,

Jay Saha

Academic Editor

PLOS ONE

Journal Requirements:

1. When submitting your revision, we need you to address these additional requirements. Please ensure that your manuscript meets PLOS ONE's style requirements, including those for file naming. The PLOS ONE style templates can be found at https://journals.plos.org/plosone/s/file?id=wjVg/PLOSOne_formatting_sample_main_body.pdf and https://journals.plos.org/plosone/s/file?id=ba62/PLOSOne_formatting_sample_title_authors_affiliations.pdf 2. Thank you for uploading your study's underlying data set. Unfortunately, the repository you have noted in your Data Availability statement does not qualify as an acceptable data repository according to PLOS's standards. At this time, please upload the minimal data set necessary to replicate your study's findings to a stable, public repository (such as figshare or Dryad) and provide us with the relevant URLs, DOIs, or accession numbers that may be used to access these data. For a list of recommended repositories and additional information on PLOS standards for data deposition, please see https://journals.plos.org/plosone/s/recommended-repositories. 3. In the online submission form, you indicated that your data will be submitted to a repository upon acceptance.  We strongly recommend all authors deposit their data before acceptance, as the process can be lengthy and hold up publication timelines. Please note that, though access restrictions are acceptable now, your entire minimal  dataset will need to be made freely accessible if your manuscript is accepted for publication. This policy applies to all data except where public deposition would breach compliance with the protocol approved by your research ethics board. If you are unable to adhere to our open data policy, please kindly revise your statement to explain your reasoning and we will seek the editor's input on an exemption.  4. Please include your full ethics statement in the ‘Methods’ section of your manuscript file. In your statement, please include the full name of the IRB or ethics committee who approved or waived your study, as well as whether or not you obtained informed written or verbal consent. If consent was waived for your study, please include this information in your statement as well. 5. We notice that your supplementary figures are uploaded with the file type 'Figure'. Please amend the file type to 'Supporting Information'. Please ensure that each Supporting Information file has a legend listed in the manuscript after the references list. 6. We notice that your supplementary figures and tables are included in the manuscript file. Please remove them and upload them with the file type 'Supporting Information'. Please ensure that each Supporting Information file has a legend listed in the manuscript after the references list. 7. Please include captions for your Supporting Information files at the end of your manuscript, and update any in-text citations to match accordingly. Please see our Supporting Information guidelines for more information: http://journals.plos.org/plosone/s/supporting-information.

Reviewers' comments:

Reviewer's Responses to Questions

**Comments to the Author**

1. Is the manuscript technically sound, and do the data support the conclusions?

Reviewer #1: Yes

Reviewer #2: Yes

2. Has the statistical analysis been performed appropriately and rigorously?

Reviewer #1: Yes

Reviewer #2: Yes

3. Have the authors made all data underlying the findings in their manuscript fully available?

Reviewer #1: Yes

Reviewer #2: Yes

4. Is the manuscript presented in an intelligible fashion and written in standard English?

Reviewer #1: Yes

Reviewer #2: Yes

Reviewer #1: Dear Editor,

This article deals with a very interesting topic. The relevance of this article is undeniable. Additionally, the article is well structured, the methods of analysis are relevant. However, before its publication, some changes and additions need to be made.

You will find my comments below.

Sincerely yours

1. In the results section, I suggest that the authors present the socio-demographic characteristics of the respondents. It allows to better understand bivariate and multivariate analysis.

2. In the discussion section, I suggest that the authors focus on India, taking into account the socio-demographic characteristics of the country's urban/rural areas. Yes, it's good to compare your results with those of other authors, but the discussion must take account of the country's socio-economic reality.

3. I suggest that the authors rephrase it ‘This is due to their increased likelihood of acculturation in the urban areas.’ the term acculturation is a bit pejorative. preferably, they have a better chance of adapting to the urban lifestyle and practices.

4. I suggest that the authors add a section ‘Policy implication of findings’.

Reviewer #2: The study addresses a critical gap in research on rural-to-urban migration and its association with obesity in the Indian context, a topic that has been underexplored compared to Western settings. The clear association between prolonged urban residence and elevated obesity risk offers valuable insights into how lifestyle and environmental changes in urban areas contribute to obesity among migrants.

I have few suggestions which is given below:

Introduction

The introduction provides a broad overview of the association between migration and health outcomes, specifically obesity. However, the structure could benefit from more logical flow. For example, after discussing the global relevance of rural-to-urban migration, the author could transition more smoothly into the context of obesity in India. Introducing the knowledge gap earlier in the paragraph would make the problem clearer upfront.

Overall, the introduction provides a solid foundation for the study, highlighting the global and Indian context of migration and obesity. However, it could be made more concise, logically organized, and focused on the specific research gap in India.

-Reorganize the introduction to first establish the global connection between migration and obesity, then introduce the Indian context and the gaps in research.

Methods

Methods have been clearly explained by the author.

Results

Results are well written with full details which will be helpful to the readers.

Discussion

-The lack of dietary data is a significant limitation that affects the interpretation of the findings. While the study acknowledges this gap, future research should incorporate dietary assessments to provide a more complete picture of how urban diets contribute to obesity. This could involve collecting detailed information on food consumption patterns and their relationship with obesity.

-The discussion on socio-economic status and its impact on obesity could be expanded.

-The observation of greater obesity risk among female migrants is an important finding. It would be helpful to explain deeper into potential reasons for this disparity, such as cultural, socio-economic, or behavioral factors that might contribute to higher obesity rates in female migrants compared to their male counterparts.

-The manuscript notes that educational attainment did not show a uniform increase in obesity risk. This point could be elaborated further.

Overall, the study provides valuable insights into the relationship between migration and obesity. Addressing these comments and suggestions could further enhance the impact and clarity of the research.

**Do you want your identity to be public for this peer review?** For information about this choice, including consent withdrawal, please see our Privacy Policy

Reviewer #1: No

Reviewer #2: No

---

## [Author Response · Author response to Decision Letter 1]

21 Oct 2024

Dear editor,

I would like to express my sincere gratitude for the thorough and insightful guidance provided during the review process of our manuscript. Your expertise and thoughtful suggestions have been instrumental in enhancing the quality and clarity of our work. We deeply appreciate the time and effort you have dedicated to ensuring our research meets the highest standards of publication.

The comments from the reviewers have significantly benefited the manuscript, and we have addressed them to the best of our capacity. We deeply appreciate the time and effort you and the reviewers have dedicated to ensuring our research meets the highest standards of publication.

Journal Requirements:

We have thoroughly reviewed and revised our manuscript to ensure it meets all the PLOS ONE style requirements. We have carefully followed the style templates provided and adhered to the file naming conventions as specified by the journal. We believe these adjustments will facilitate a smooth review and publication process. Thank you for guiding us to make these necessary changes.

At this time, please upload the minimal data set necessary to replicate your study's findings to a stable, public repository (such as figshare or Dryad) and provide us with the relevant URLs, DOIs, or accession numbers that may be used to access these data. For a list of recommended repositories and additional information on PLOS standards for data deposition, please

see https://journals.plos.org/plosone/s/recommended-repositories.

A proper repository of the dataset has now been added. We also shared the replicable data through figshare.

3. In the online submission form, you indicated that your data will be submitted to a repository upon acceptance. We strongly recommend all authors deposit their data before acceptance, as the process can be lengthy and hold up publication timelines. Please note that, though access restrictions are acceptable now, your entire minimal dataset will need to be made freely accessible if your manuscript is accepted for publication. This policy applies to all data except where public deposition would breach compliance with the protocol approved by your research ethics board. If you are unable to adhere to our open data policy, please kindly revise your statement to explain your reasoning and we will seek the editor's input on an exemption.

The data is already available in a public repository. We believe, earlier there was a mistake on our part. We now confirm the accessibility of the data through the publicly available repository

Done!

5. We notice that your supplementary figures are uploaded with the file type 'Figure'. Please amend the file type to 'Supporting Information'. Please ensure that each Supporting Information file has a legend listed in the manuscript after the references list.

Done!

6. We notice that your supplementary figures and tables are included in the manuscript file. Please remove them and upload them with the file type 'Supporting Information'. Please ensure that each Supporting Information file has a legend listed in the manuscript after the references list.

Done!

Done!

Done!

Comments to the Author

Reviewer 1

This article deals with a very interesting topic. The relevance of this article is undeniable. Additionally, the article is well structured, the methods of analysis are relevant. However, before its publication, some changes and additions need to be made.

You will find my comments below.

Thank you for your thoughtful and constructive feedback on our manuscript. We appreciate your recognition of the relevance and structure of our article, as well as the validation of our analytical methods. We are committed to addressing the changes and additions you have suggested to enhance the quality and impact of our work.

1. In the results section, I suggest that the authors present the socio-demographic characteristics of the respondents. It allows to better understand bivariate and multivariate analysis.

Table of the socio-demographic characteristics of the respondents has been now moved from supplementary to the main manuscript. We briefly discussed the characteristics of the individuals.

2. In the discussion section, I suggest that the authors focus on India, taking into account the socio-demographic characteristics of the country's urban/rural areas. Yes, it's good to compare your results with those of other authors, but the discussion must take account of the country's socio-economic reality.

Done positively.

3. I suggest that the authors rephrase it ‘This is due to their increased likelihood of acculturation in the urban areas.’ the term acculturation is a bit pejorative. preferably, they have a better chance of adapting to the urban lifestyle and practices.

Done!

4. I suggest that the authors add a section ‘Policy implication of findings’.

A new section has now been added before the conclusion, following your instruction.

Reviewer 2

The study addresses a critical gap in research on rural-to-urban migration and its association with obesity in the Indian context, a topic that has been underexplored compared to Western settings. The clear association between prolonged urban residence and elevated obesity risk offers valuable insights into how lifestyle and environmental changes in urban areas contribute to obesity among migrants.

Thank you for your encouraging comments and for recognizing the importance of our study addressing the critical gap in research on rural-to-urban migration and its association with obesity in the Indian context. We are grateful for your acknowledgment of the value our findings.

I have few suggestions which is given below:

Introduction

The introduction provides a broad overview of the association between migration and health outcomes, specifically obesity. However, the structure could benefit from more logical flow. For example, after discussing the global relevance of rural-to-urban migration, the author could transition more smoothly into the context of obesity in India. Introducing the knowledge gap earlier in the paragraph would make the problem clearer upfront.

Overall, the introduction provides a solid foundation for the study, highlighting the global and Indian context of migration and obesity. However, it could be made more concise, logically organized, and focused on the specific research gap in India.-Reorganize the introduction to first establish the global connection between migration and obesity, then introduce the Indian context and the gaps in research.

Thank you for your constructive critique regarding the structure of our introduction. We have considered your feedback and revised it accordingly, enhancing the logical flow by clearly linking the global context of migration to the specific issues of obesity in India early in the text.

Methods

Methods have been clearly explained by the author.

Results

Results are well written with full details which will be helpful to the readers.

Discussion

-The lack of dietary data is a significant limitation that affects the interpretation of the findings. While the study acknowledges this gap, future research should incorporate dietary assessments to provide a more complete picture of how urban diets contribute to obesity. This could involve collecting detailed information on food consumption patterns and their relationship with obesity.

The discussion on socio-economic status and its impact on obesity could be expanded.

We have addressed this positively.

The observation of greater obesity risk among female migrants is an important finding. It would be helpful to explain deeper into potential reasons for this disparity, such as cultural, socio-economic, or behavioral factors that might contribute to higher obesity rates in female migrants compared to their male counterparts.

Explanation has been added following your suggestion.

The manuscript notes that educational attainment did not show a uniform increase in obesity risk. This point could be elaborated further. Overall, the study provides valuable insights into the relationship between migration and obesity. Addressing these comments and suggestions could further enhance the impact and clarity of the research.

Done!

---

## [Editor Report · Decision Letter 1]

Dear Dr. Mandal,

Thank you for submitting your manuscript to PLOS ONE. After careful consideration, we feel that it has merit but does not fully meet PLOS ONE’s publication criteria as it currently stands. Therefore, we invite you to submit a revised version of the manuscript that addresses the points raised during the review process.

We look forward to receiving your revised manuscript.

Kind regards,

Jay Saha

Academic Editor

PLOS ONE
---

## [Author Response · Author response to Decision Letter 2]

26 Dec 2024

Dear Editor

Thank you for providing the opportunity to revise and resubmit our manuscript

We have carefully addressed all the comments and suggestions provided by the academic editor and reviewers. Following your instructions, we have uploaded the following documents:

Response to Reviewers: A detailed rebuttal letter addressing each point raised by the academic editor and reviewers.

Revised Manuscript with Track Changes: A marked-up version of the manuscript highlighting all the changes made to the original submission.

Manuscript: A clean, unmarked version of the revised paper without tracked changes.

We have ensured that the revisions align with the feedback provided and believe that the updated manuscript is stronger and more comprehensive as a result of this process.

---

## [Decision Letter · Decision Letter 2]

Dear Dr. Mandal,

Thank you for submitting your manuscript to PLOS ONE. After careful consideration, we feel that it has merit but does not fully meet PLOS ONE’s publication criteria as it currently stands. Therefore, we invite you to submit a revised version of the manuscript that addresses the points raised during the review process.

We look forward to receiving your revised manuscript.

Kind regards,

Hansani Madushika Abeywickrama, Ph.D.

Academic Editor

PLOS ONE

Journal Requirements:

Reviewers' comments:

Reviewer's Responses to Questions

**Comments to the Author**

Reviewer #3: (No Response)

Reviewer #4: (No Response)

2. Is the manuscript technically sound, and do the data support the conclusions?

Reviewer #3: Yes

Reviewer #4: Yes

3. Has the statistical analysis been performed appropriately and rigorously?

Reviewer #3: Yes

Reviewer #4: Yes

4. Have the authors made all data underlying the findings in their manuscript fully available?

Reviewer #3: Yes

Reviewer #4: Yes

5. Is the manuscript presented in an intelligible fashion and written in standard English?

Reviewer #3: Yes

Reviewer #4: Yes

Reviewer #3: Technical Soundness and Data Support for Conclusions

Strengths: The study is well-designed, using a nationally representative dataset (LASI) and robust methods (logistic/quantile regression). The conclusions align with global literature on migration and obesity.

Limitations: The cross-sectional design limits causal inference. Lack of dietary data is noted, but the authors justify this by referencing prior studies on urban dietary shifts. Statistical Analysis

Appropriateness: The use of logistic regression (for binary outcomes) and quantile regression (to explore BMI distribution) is rigorous. Multicollinearity checks (VIF < 2) and sensitivity analyses (waist circumference, waist-hip ratio) strengthen validity.

Suggestions: Clarify why certain covariates (e.g., alcohol/tobacco) showed inconsistent associations. Additional robustness checks (e.g., stratified analyses by gender/age) could enhance insights. Data Availability

Compliance: Uploaded replicable data to figshare (DOI: 10.6084/m9.figshare.27266337.v1). The primary dataset is publicly available via the Gateway to Global Aging Data (www.g2aging.org).

Final Assessment

The manuscript is technically sound, with conclusions well-supported by data.

Statistical methods are appropriate and rigorous, though minor clarifications could improve transparency.

Data availability meets PLOS ONE standards after revision.

The Conclusion section of the manuscript does not clearly and directly address the study's stated objectives. Here’s a breakdown of the issues and suggestions for improvement: Key Problems with the Current Conclusion

Lacks Direct Link to Objectives

The study aimed to:

Examine the impact of rural-to-urban migration on obesity.

Compare obesity risk by duration of urban residence.

The conclusion vaguely summarizes findings but does not explicitly tie them back to these objectives.

Too General and Repetitive

It reiterates results (e.g., "prolonged urban residence is progressively associated with obesity risks") without synthesizing implications.

Reviewer #4: The manuscript provides a comprehensive analysis of the association between rural-to-urban migration and obesity among middle-aged and older adults in India. The study is well-structured, methodologically sound, and addresses an important public health issue. However, some areas require clarification, refinement, or expansion to enhance the manuscript's impact and readability:

- Clarify the Research Gap: The introduction highlights the lack of nationally representative studies on migration and obesity in India but could better emphasize the novelty of focusing on middle-aged and older adults.

- Migration Definition: Line 143-146: Based on the definition, it seem like even participants who just moved this area for 1 day were also categorized as migrant. Is there any reference to support this?

Please verify some typo errors:

- Line 380: Socio-economic stata  Socioeconomic status

**Do you want your identity to be public for this peer review?** For information about this choice, including consent withdrawal, please see our Privacy Policy

Reviewer #3: **Yes: ** Ananda Chandrasekara

Reviewer #4: No

---

## [Decision Letter · Decision Letter 3]

Understanding the impact of urban exposure on obesity among middle and old-age migrants in India

PONE-D-24-28968R3

Dear Dr. Mandal,

We’re pleased to inform you that your manuscript has been judged scientifically suitable for publication and will be formally accepted for publication once it meets all outstanding technical requirements.

Kind regards,

Hansani Madushika Abeywickrama, Ph.D.

Academic Editor

PLOS ONE

Additional Editor Comments (optional):

I would like to commend the authors on a well-structured and clearly written manuscript.

Reviewers' comments:

Reviewer's Responses to Questions

**Comments to the Author**

Reviewer #3: All comments have been addressed

2. Is the manuscript technically sound, and do the data support the conclusions?

Reviewer #3: Yes

3. Has the statistical analysis been performed appropriately and rigorously?

Reviewer #3: Yes

4. Have the authors made all data underlying the findings in their manuscript fully available?

Reviewer #3: Yes

5. Is the manuscript presented in an intelligible fashion and written in standard English?

Reviewer #3: Yes

Reviewer #3: Strengths:

The manuscript is structured logically, including clearly defined sections such as Abstract, Introduction, Methods, Results, Discussion, and Conclusion.

The language used throughout is predominantly clear, formal, and consistent with academic standards.

Technical terms and statistical methods (logistic regression, quantile regression, BMI, waist circumference, etc.) are described adequately and appropriately.

Areas for Improvement:

Although the manuscript is broadly well-written, there are several issues related to language, clarity, grammar, and style that need addressing:

Clarity and Precision:

The Abstract is clear but can be more concise and direct. For instance, clearly stating the implications or recommendations briefly would enhance reader engagement.

The Introduction section effectively provides context, but some sentences could be more precise, particularly regarding the exact link between migration and obesity in the Indian context.

Grammar and Syntax:

A few sentences are lengthy and occasionally awkwardly constructed, making comprehension slightly challenging at points. Breaking these sentences into shorter, clearer phrases could improve readability significantly.

Examples of sentences needing grammatical refinement:

Original: "However, whether migration to urban areas ages has any impact on obesity in India is inconclusive and scarce."

Improved: "However, evidence regarding the impact of rural-to-urban migration on obesity among older adults in India remains inconclusive and limited."

Original: "To fulfil the study objective, this study employed logistic and quantile regression techniques."

Improved: "Logistic and quantile regression analyses were employed to fulfill the study objective."

Terminology and Definitions:

Consistent and precise definitions of key concepts (e.g., clear distinction between obesity, overweight, abdominal obesity) should be consistently maintained throughout.

Consistency:

The manuscript occasionally switches between past and present tense. Maintaining consistent tense usage, especially within the Methods and Results sections, is important for readability and academic rigor.

Punctuation and Formatting:

Some minor punctuation inconsistencies (e.g., misplaced commas, semicolons, or spacing issues) should be addressed.

Consistent formatting of tables and figures, with clear and detailed captions, would enhance intelligibility.

**Do you want your identity to be public for this peer review?** For information about this choice, including consent withdrawal, please see our Privacy Policy

Reviewer #3: **Yes: ** Prof. Ananda Chandrasekara

---

## [Editor Report · Acceptance letter]

PONE-D-24-28968R3

PLOS ONE

Dear Dr. Mandal,

I'm pleased to inform you that your manuscript has been deemed suitable for publication in PLOS ONE. Congratulations! Your manuscript is now being handed over to our production team.

Kind regards,

on behalf of

Dr. Hansani Madushika Abeywickrama

Academic Editor

PLOS ONE